# The Oxidative and Color Stability of Beef from Steers Fed Pasture or Concentrate during Retail Display

**DOI:** 10.3390/ani13182972

**Published:** 2023-09-20

**Authors:** Alejandra Terevinto, María Cristina Cabrera, Fernanda Zaccari, Ali Saadoun

**Affiliations:** 1Facultad de Agronomía, Universidad de la República, Av. Garzón 780, Montevideo 12900, Uruguay; mcab@fagro.edu.uy (M.C.C.); fzaccari@fagro.edu.uy (F.Z.); asaadoun@fcien.edu.uy (A.S.); 2Facultad de Ciencias, Universidad de la República, Calle Iguá 4225, Montevideo 11400, Uruguay

**Keywords:** lipid oxidation, protein oxidation, β-carotene, α-tocopherol, fatty acid, heme iron, beef, pasture, concentrate

## Abstract

**Simple Summary:**

In Uruguay and some other countries in the region, where pasture grazing has always been predominant in beef production, steers are now increasingly fed grains to shorten the production time. This has implications for meat quality, so meat from both systems (pasture and concentrate) was compared under refrigerated retail conditions in this study. Meat from pasture-fed steers exhibited lower levels of lipid and protein oxidation during the display period, likely due to the higher levels of antioxidants, such as β-carotene and α-tocopherol, found in this investigation. In addition, meat from pasture-raised steers was healthier for consumers in terms of its fatty acid composition and lower intramuscular fat content. In conclusion, meat from pasture-fed steers was more stable during retail refrigeration, possibly allowing for a longer shelf-life, and was healthier for consumers compared to meat from concentrate-fed steers.

**Abstract:**

Beef production in Uruguay is based on pasture (~85%) or concentrate (~15%), resulting in differences in meat quality. The objective of this study was to compare the oxidative stability and color of beef from these two systems during refrigerated retail display. For these purposes, the *Semimembranosus* muscle was removed from ten Aberdeen Angus steers raised and fed on pasture (130 days prior to slaughter) and from another ten steers fed concentrate (100 days prior to slaughter), sliced. The muscles were placed in a refrigerated showcase for 3, 6, and 9 days. The contents of β-carotene, α-tocopherol, and fatty acids were determined before the meat was placed on display. Lipid and protein oxidation, color, and heme iron content were determined before and during display. The meat from pasture-fed steers had a lower intramuscular fat content (1.78 ± 0.15 vs. 4.52 ± 0.46), lower levels of monounsaturated fatty acids, a lower n-6/n-3 ratio, less lipid and protein oxidation, lower L* and a* values, and higher levels of α-linolenic acid, DHA, total n-3, β-carotene, and α-tocopherol. In conclusion, the meat from pasture-fed steers was more stable during retail display from an oxidative point of view, which may be due to its higher levels of antioxidant compounds such as β-carotene and α-tocopherol and had a healthier fatty acid profile for consumers.

## 1. Introduction

Oxidative processes are major non-microbiological factors involved in the deterioration of the quality of meat during refrigerated storage [1]. Indeed, oxidation induces modifications in muscle lipids and proteins and therefore affects the organoleptic and nutritional properties of meat and meat products. The oxidation of myoglobin and lipids reduces the color and flavor acceptability of fresh meat during refrigerated storage [2,3]. In fact, color is an important parameter for consumers that has a substantial influence on acceptability and purchasing decisions at retail points [4]; a bright red color in beef is indicative of freshness [5]. When lipid peroxidation propagates in membranes, it promotes the oxidation of myoglobin, resulting in color deterioration and the formation of rancid odors and other off flavors in fresh meat [6]. This is why the beef industry is concerned with the shelf life of products [7]. The stability of meat with respect to oxidation is the result of a balance between prooxidants and antioxidants [1]. Transition metals, specifically iron, are considered strong pro-oxidant species [8]. Dietary antioxidants can be delivered to the muscle where, together with endogenous defense systems, they counteract the action of prooxidants [9]. However, endogenous antioxidant concentrations depend on the animal species, muscle type, and diet [10,11,12]. As shown in other works [9,13,14], dietary treatments may directly affect the balance between antioxidant and prooxidant components in muscle.

The beef cattle production systems in Uruguay rely almost exclusively (~85%) on grazed pastures. However, more recently, intensive beef production systems have received an increased amount of interest from some beef producers. These two systems produce meat with different antioxidant, prooxidant, and fatty acid contents and compositions [9,15,16]. Fresh herbage is rich in antioxidants such as tocopherols, carotenoids, ascorbic acid, and phenolic compounds [12], and grains contain polyphenols like proanthocyanidins and phytic acid [17]. Indeed, tocopherols and β-carotene constitute the main lipid-soluble free radical scavengers in meat. Tocopherols are a group of plant phenolic isomers deposited from dietary sources [18], and several studies indicate that α-tocopherol delays the oxidation of myoglobin and extends color stability in retail beef [19,20,21]. In addition, β-carotene quenches sites localized within the hydrophobic regions of biological membranes in contrast with the scavenging activity of α-tocopherol, which operates close to the membrane’s surface [22].

Thus, the aim of this investigation was to study the oxidative and color stability of beef during refrigerated retail display, simulating typical commercial conditions and assessing the α-tocopherol, β-carotene, and fatty acid contents prior to the display, utilizing meat from animals produced in pastures and in concentrate finishing systems.

## 2. Materials and Methods

### 2.1. Muscle Samples

The *Semimembranosus* muscle was selected from twenty young Aberdeen Angus steers (two–teeth) aged from 26 to 30 months on the same day in a commercial slaughterhouse named Frigorífico Breeders & Packers of Uruguay (BPU), which is located in the city of Durazno. All the steers were raised and sacrificed according to the good animal welfare practices approved by the Honorary Committee for Animal Experimentation (CHEA, protocol ID 702, approved on 14 February 2020) and came from commercial operations run by the Breeders Society of Aberdeen Angus of Uruguay (SCAAU).

Ten animals with an average final live weight of 495.8 kg were reared under characteristic Uruguayan conditions based on the exploitation of natural resources with traditional extensive grazing. They grazed on natural pastures (gramineae and legumes) and were finished (130 days before slaughter) on an improved pasture (comprising 40% natural and 60% cultivated grass) consisting of tall fescue (*Festuca arundinacea*), white clover (*Trifolium repens*), and birdsfoot trefoil (*Lotus subbiflorus* cv. El Rincón).

The other ten animals, with an average final live weight of 498.2 kg, came from an intensive production farm that was registered and authorized and which followed the requirements for exportation to the European Union (EU), which are stated in the European Commission Regulation (Number 481/2012). Quota 481 requires that at least 100 days prior to slaughter, animals receive the concentrate (which comprises 62% concentrated or cereal coproducts in the form of dry matter, at least 12.26 MJ/kg of dry matter, and at least 1.4% of body weight in a dry-matter-based form). The diet usually consists of whole plant sorghum silage (comprising 34% dry matter), wet grain sorghum (74% dry matter) and wet grain corn silage (70% dry matter), sunflower pellets (26% protein), and a mineral–vitamin premix, urea, and ionophores. A complete diet contains approximately 12% crude protein. The animals were expected to consume about 15.7 kg/animal/day of the complete diet (10.4 kg of dry matter/animal/day). 

According to the official classification system of INACUR-Uruguay, carcass conformation and fatness were classified as A and 2, respectively, for both feeding systems [23]. The carcasses were kept refrigerated at 1–2 °C for 36 h post mortem, and the *Semimembranosus* muscle was then sampled, vacuum-packaged, and maintained at 1–2 °C in the meat processing plant. This muscle forms part of a cut (inside round) that is commonly seen in the form of steaks in refrigerated showcases at retail points and is highly consumed in Uruguay because it is lean and more affordable for consumers. Once in the laboratory, the muscle samples were cut into slices which weighed 100 g and were 1 cm thick; the color of the samples was tested, and some of the samples (day 0) were immediately frozen at −18 °C until further determinations were made. The rest of the samples were placed on food-grade polyfoam trays and wrapped with a food-grade, oxygen-permeable polyvinyl chloride film (density: 1.39 g/cm^3^; O_2_ permeability: cm^3^ mm/m^2^ day atm). Trays containing 3 slices of meat from each animal were placed in a commercial refrigerated showcase (CE, SS1500 model; 1.25 m tall, 90 cm wide, and 1.50 m long) at 2–8 °C with artificial light (120 cm, 18 W, 2700 Kelvin, and 1700 lm). 

After each display time was completed (3, 6, and 9 days), a sample was tested for color, immediately vacuum-packaged (105 µm, Lacor Menaje Profesional S.L., LR69454, Bergara, Spain), and frozen at −18 °C until further determinations were made (n = 10 for each feeding system and display time). 

### 2.2. Determination of Lipid Oxidation 

The degree of lipid oxidation in the meat was determined on day 0 and after 3, 6, and 9 days of display. Samples of 5 g of frozen meat were homogenized in a Waring blender with 100 mL of extraction buffer (containing 0.15 M of KCl, 0.02 M of EDTA, and 0.30 M of BHT) at 12,000 rpm for 1 min. Part of the homogenate was frozen for carbonyl and protein content assays, and part was used for the TBARS (thiobarbituric acid reactive species) test. The TBARS procedure for the determination of lipid oxidation was followed according to [24,25], as described in [15]. The results were expressed as mg of malondialdehyde/kg of fresh meat.

### 2.3. Determination of Protein Oxidation

The level of protein oxidation was determined in the meat on day 0 and after 3, 6, and 9 days of display, following the carbonyl protein assay according to [26], as described in [15]. The results were expressed as nmoles of dinitrophenylhydrazine/mg of protein. The protein content was determined at 280 nm in the extraction buffer using bovine serum albumin (BSA) from Sigma chemicals Co. (St. Louis, MI, USA) as a protein standard, as described in [27].

### 2.4. Β-Carotene and α-Tocopherol Contents

Both compounds were extracted from the meat on day 0, following [28], Ref. [18] for carotenoids, and [29] for tocopherol, with some modifications. Briefly, 0.5 g of frozen meat was homogenized with 11 mL of methanol: THF (1:1) in an Ultra-Turrax IKA T18 basic for 30 s at 12,000 rpm. Then, centrifugation was carried out at 12,000 rpm and 4 °C for 1 min. The supernatant was kept at −80 °C, and prior to measurement, an aliquot of 1 mL was taken and evaporated using N_2_. The pellet was resuspended in 0.2 mL of the same extraction solution, and 20 μL was injected into the HPLC (Thermo Fisher Scientific Inc., Thermo Separation Products, Spectral Series P100, Waltham, MA, USA) with a UV2000 detector. The β-carotene content was measured at 450 nm and the α-tocopherol content was measured at 290 nm via a C30 thermostatted column at 30 °C. The mobile phase used was ethanol: methanol: THF (75:20:5), HPLC grade, with a flux of 0.5 mL/min. Calibration curves were produced using β-carotene (Sigma Aldrich Co., C9750, St. Louis, MO, USA) and DL-α-tocopherol (Sigma Aldrich Co., T3251, St. Louis, MO, USA) standards. The results were expressed as μg/g fresh meat. 

### 2.5. Determination of Fat Content and Fatty Acid Composition

The total intramuscular fat was extracted from the meat on day 0, following the chloroform: methanol (2:1; *v*:*v*) procedure described in [30]. An extract containing approximately 40 mg of fat was methylated following the method described in [31], and a fatty acid analysis was performed via gas chromatography, using a Clarus 500 (Perkin Elmer Instruments Inc., Shelton, CT, USA) split/splitless instrument equipped with a 100 m CPSIL 88 capillary column. Hydrogen was the carrier gas (flow rate: 1 mL/min), and an FID detector was used. Fatty acids (FAMEs) were determined by comparing the retention times to fatty acid standards. 

### 2.6. Color Measurement

Color parameters were measured in the meat on day 0 and after 3, 6, and 9 days of display, using a Minolta CR-10 colorimeter (Konica Minolta Inc., Tokyo, Japan) with a D65 standard illuminant, a observer angle of 10°, and an 8 mm aperture. The CIE L*a*b* system provided the values of three color components: L* (0: black to 100: white component; lightness), a* (+ red to − green component), and b* (+ yellow to − blue component) [32]. All the samples were allowed to bloom for 10 min before the measurements were taken. The hue angle (tone) was calculated as tan^−1^ (b*/a*), and chroma (saturation) was calculated as [(a*)^2^ + (b*)^2^]^1/2^.

### 2.7. Heme Iron Content

The total heme pigments in the meat were determined on day 0 and after 3, 6, and 9 days of display as hemin after their extraction with an acidified acetone solution, as described in [33,34,35]. The heme iron content was calculated using the factor 0.0882 μg of iron/μg of hematin [33], and the results were expressed as ppm. All samples were assayed in duplicate.

### 2.8. Statistical Analysis

The data regarding the TBARS, carbonyls, β-carotene, α-tocopherol, total fat content, fatty acid composition, color parameters, and heme iron content in the meat were reported as means ± the standard error of the mean (SEM). The experimental unit was muscle (coming from ten animals in each system, n = 10). Data regarding lipid and protein oxidation, color, and heme iron content were analyzed using a repeated-measures ANOVA, taking the feeding system and display days as fixed effects. In addition, a one-way ANOVA followed by the Tukey–Kramer multiple comparisons test was used to compare feeding the systems on each day of the display. A one-way ANOVA followed by the Tukey–Kramer multiple comparisons was also used to compare the days of display in each feeding system. Data regarding the intramuscular fat, fatty acid, α-tocopherol, and β-carotene contents were analyzed using a one-way ANOVA. The statistical analysis was performed using NCSS 2007 software (NCSS, 329 North 1000 East, Kaysville, UT, USA), and the level of significance was established at *p* < 0.05.

## 3. Results

### 3.1. Lipid and Protein Oxidation

The levels of lipid oxidation and protein oxidation were lower (main effect of *p* < 0.0001 and *p* < 0.01, respectively) in meat from pasture-fed Aberdeen Angus steers compared with meat from concentrate-fed Aberdeen Angus steers (Figure 1).

When the display day effect was considered, an increment in both lipid and protein oxidation (*p* < 0.0001 for both) with the time of the display was found. When the evolution of lipid oxidation was analyzed day by day (Figure 1a), a significant increment in TBARS can be seen at the beginning of the display (day 0 vs. day 3) in meat from concentrate-fed animals and at days 6 and 9 in meat from pasture-fed animals. In addition, the carbonyl content (Figure 1b) increased significantly from day 3 to day 6, from day 6 to day 9 of the display in the meat from concentrate-fed animals, and from day 6 to day 9 in the meat from pasture-fed animals.

### 3.2. β-Carotene and α-Tocopherol Contents

As observed in Figure 2, the α-tocopherol content in the meat measured before the display was almost twofold greater (*p* < 0.05) in the meat from pasture-fed animals (3.7 ± 0.5 μg/g fresh meat) than in the meat from concentrate-fed animals (1.9 ± 0.3 μg/g fresh meat). Regarding the β-carotene content measured before the display, it was six times higher (1.73 ± 0.18 μg/g fresh meat) in the meat from pasture-fed Aberdeen Angus steers than in the meat from concentrate-fed steers (0.27 ± 0.04 μg/g fresh meat) (*p* < 0.0001).

### 3.3. Lipid Content and Fatty Acid Composition

The meat from pasture-fed Aberdeen Angus steers presented a lower intramuscular fat content (Table 1) than the meat from concentrate-fed Aberdeen Angus steers (*p* < 0.0001).

Regarding the fatty acid composition of the intramuscular fat measured prior to the display, no differences between feeding systems were found in the even-chain saturated fatty acids (SFAs) 12:0, 14:0, 16:0, and 18:0, nor in the MUFAs 14:1, 16:1, 17:1, 18:1, and 20:1. The contents of the n-6 PUFAs 18:2 and 20:4, the n-3 PUFAs EPA and DPA, and CLA in the intramuscular fat were also not significantly different between feeding systems (Table 1). However, higher contents of α-linolenic acid (C18:3), C20:3, and DHA, which are n-3 PUFAs, were observed in the meat from pasture-fed animals compared with the meat from concentrate-fed animals (Table 1). In particular, the α-linolenic acid content was 3.4-fold greater, and the DHA content was 5-fold greater. 

Regarding the total SFA, total PUFA, and total n-6 contents, no differences between feeding systems were observed (Table 1). Despite this, the total MUFA content was lower (*p* < 0.05), and the total n-3 content was higher (*p* < 0.001) in the meat from the pasture systems. In addition, the n-6/n-3 ratio was lower (*p* < 0.05) in the meat from pasture-fed steers (Table 1). No differences were found between feeding systems in the PUFA/SFA ratio. 

### 3.4. Color

When evaluating the main effects for the color parameters, a feeding system effect was found for L* (*p* < 0.01) and a* (*p* < 0.05) in which the meat from pasture-fed animals showed lower values (Figure 3). This translates into darker and less red meat compared with the concentrate-produced meat. However, no feeding system effect was found for the b* parameter (Figure 3c) in the present study.

When the display day effect was evaluated, an effect was found for the three parameters (*p* < 0.0001 in all cases) (Figure 3). The L* value was higher on day 9 compared with the other days of the display, so the meat became brighter with time (Figure 3a). In meat from pasture-fed steers, L* values were relatively constant; only a significant increase was observed from day 3 to day 6. In the meat from concentrate-fed steers, L* values decreased from day 0 until day 6 and then increased until day 9. When observing L* values on each day of the display, differences between feeding systems can be seen at the beginning (day 0) and at the end of the display (day 9) in which the meat from concentrate-fed animals was brighter. In addition to this, a decrease in the a* value was observed (Figure 3b) in meat from both systems, so the meat lost redness with days on display, and the b* value was higher at the beginning and at the end of the display (Figure 3c).

Associated with the a* and b* parameters are the hue (°) and chroma values. In the present study, a feeding system effect was observed for hue (*p* < 0.05) in which the meat from pasture-fed animals had a higher tone value than the meat from concentrate-fed animals (Figure 4a). A display day effect was observed for hue (*p* < 0.0001) in which the values were lower on days 0 and 3 than on day 6, which were lower than on day 9. A significant increase in the hue value was observed on day 3 in the meat from pasture-fed steers and on day 6 in the meat from concentrate-fed steers. On day 6, we can see that the tone is significantly different between the meats (*p* < 0.05, pasture > feedlot). 

No feeding system effect was found for chroma (saturation), but a display day effect was observed (*p* < 0.0001). The color saturation diminished from the fresh meat (day 0) until day 6 of display and then remained steady until the end (day 9). As can be seen in Figure 4b, the color saturation diminishes with the length of display time and at the same rate in the meat from concentrate- and pasture-fed steers. 

### 3.5. Heme Iron Content

The heme iron content results did not show any statistical difference between the feeding systems (pasture vs. concentrate) or between the days of the display (Figure 5). 

## 4. Discussion

The results found for lipid oxidation in the current study, in which the meat from pasture-fed Aberdeen Angus steers presented lower levels of TBARS compared with the meat from concentrate-fed Aberdeen Angus steers during refrigerated storage, agreed with the results of other publications. For example, the results in Ref. [36], in which differences in strip loins from Angus and Angus–Hereford cross steers were assessed between conventional grain-finished beef and beef fed with grass for 25 months, and the results in Ref. [37], in which both loin steaks from Tudanca bulls were assessed over 6 days of refrigerated retail display and [8] the *Longissimus thoracis* muscle from British and Zebu cross steers were assessed over 13 days of refrigerated storage. There are a few studies concerning protein oxidation in meat, and our results agree with the results in [4] in which *Psoas major* steaks from crossbreed steers were assessed over 9 days in retail display and meat from pasture-fed animals presented lower levels of protein carbonyls compared with meat from concentrate-fed animals. All these results indicate that beef coming from pasture systems has greater oxidative stability under the conditions of refrigerated retail display than beef coming from concentrate feeding systems. This can be explained by the higher contents of α-tocopherol and β-carotene found in the meat before the refrigerated retail display in the present investigation. It is known that α-tocopherol and β-carotene provide antioxidant protection in meat, as observed in previous investigations [38,39]. Vitamin E (α-tocopherol) protects cells against the effects of free radicals and may also block the formation of nitrosamines [19]. In addition, carotenes can quench singlet oxygen and scavenge toxic free radicals, preventing or reducing damage to living cells. They can also react chemically with free radicals, and the system of conjugated double bonds is directly destroyed. Due to their long conjugated chains, carotenoids are highly reactive [40]. As the PUFA contents did not differ between the feeding systems, they could not explain the higher levels of lipid oxidation in the meat from concentrate-fed animals due to the greater contents of PUFAs. It is relevant to mention that the lipids containing PUFAs are particularly susceptible to free radical attack. It is also important to consider that the meat from concentrate-fed steers had a higher intramuscular fat content, which may have caused greater levels of lipid oxidation in the meat. 

The increments observed in lipid and protein oxidation with the display time were in accordance with other research [4,41,42,43] and could be explained by the presence of oxygen in the air and the artificial light of the showcase, which favor oxidative processes [44]. However, despite the increment of lipid oxidation during the time of display, the maximum TBARS values reached in the meat from both feeding systems are lower than those proposed as a threshold of acceptability by consumers, 2 mg MDA/kg meat [45]. Following the results obtained for lipid and protein oxidation day by day during the retail display in both feeding systems, the oxidation processes started to increase earlier in the meat from concentrate-fed Aberdeen Angus steers than in the meat from pasture-fed Aberdeen Angus steers. A similar pattern was reported in [37] in loin steaks from Tudanca bulls over 6 days in a refrigerated retail display and in [4] for *Psoas major* steaks from crossbreed steers over 9 days in a refrigerated retail display. These results could be explained by the higher levels of antioxidant protection from α-tocopherol and β-carotene present in the meat from pasture-fed animals, which can delay the oxidation processes in this meat.

Furthermore, the higher content of β-carotene found in the meat from pasture-fed steers agrees with other publications [4,46,47,48] and may occur due to higher concentrations of β-carotene in pastures, legumes, and other green plants, while seeds and whole-plant silage have low vitamin contents [49]. In addition, the higher content of α-tocopherol found in the meat from pasture-fed steers is supported by numerous publications [4,8,18,37,46,48,50,51] and can be explained by the greater concentration of α-tocopherol in green leaf tissues compared with grains [26]. As was mentioned in [52], desirable levels of α-tocopherol in muscle must be above 3.5 μg/g to minimize lipid oxidation processes in meat. Therefore, this threshold level was exceeded in the meat from pasture-fed steers (3.7 ± 0.5 µg/g fresh meat) but was not reached in the meat from concentrate-fed steers (1.9 ± 0.3 µg/g fresh meat). It is important to note that this is the first time that β-carotene has been measured in meat produced in Uruguay and the second time α-tocopherol has been measured in meat produced in Uruguay. 

When observing the intramuscular fat content results, it is logical to think that the animals raised in an extensive system with large surfaces of pasture where they can move freely would deposit lower amounts of intramuscular fat in contrast to animals raised in intensive systems, which confine the animals to a limited area [19]. There are several studies [15,16,37,46,48,53,54,55] that are in accordance with our result. 

Regarding the fatty acid composition measured in the meat prior to its display, we can highlight higher contents of α-linolenic acid and DHA in the meat from pasture-fed animals on pasture because it they have positive health aspects for meat consumers as both fatty acids have antithrombogenic properties and can prevent cardiovascular diseases [56]. Similar results regarding the n-3 content in the intramuscular fat of beef were found in other national [15,16,51] and international [57] research studies. As DHA is the result of elongase activity, in the ruminant that receives n-3, it is logical to expect a higher content of DHA in meat from pasture-fed animals. It is likely that elongase enzymes with n-3 substrates are more active in ruminants in a pasture system, which justifies the formation of DHA from α-linolenic acid. These elongases may have been stimulated by clover-based pastures [58].

Despite not finding differences in PUFA contents in the present investigation, higher concentrations of PUFAs were observed in pasture-fed animals compared to grain-finished animals in several studies [46,50,51,59,60]. Concerning the PUFA/SFA ratio in meat from both feeding systems in the present study (0.10 ± 0.02 in pasture and 0.09 ± 0.02 in concentrate), both were under the recommendations (0.45) from [61]. The PUFA/SFA ratio is nowadays recommended to be above 0.4 or 0.5 to prevent an excess of SFA, which has a negative effect on the level of plasmatic LDL cholesterol, and an excess of PUFAs, some of which are precursors of clotting agents and are involved in the etiology of some cancers [62]. In addition, the higher contents of total MUFAs found in the meat from the concentrate system is in accordance with other publications [7,46,51,59,63,64] and could be due to the incidence of ingredients in the concentrates, which provide different composition profiles and are of interest for human nutrition.

Furthermore, the greater n-3 contents and the lower n-6/n-3 ratio found in the meat from the pasture-fed steers pasture are desirable because the balance between n-6 and n-3 PUFAs is an important determinant in decreasing the risk of cardiovascular diseases and in the prevention of atherosclerosis [65]. Indeed, pastures such as gramineae and legumes contain mainly n-3 fatty acids and in general, pasture-fed animals tend to accumulate more n-3 fatty acids [19,37], while the concentrate, which contained wet grain sorghum, wet grain corn silage, and sunflower pellets in the present investigation, all of which are cereals and asteracea, contain mainly n-6 fatty acids. These results were also observed in other research studies [46,60,66,67]. 

Indeed, the value observed for the n-6/n-3 ratio in the meat from the concentrate-fed steers (11.18 ± 3.96) was higher than the recommendations (4/1 or 5/1) in [61] and exceeded a value of 10, which is not recommended for the consumer’s health. Concerning the lipid content and fatty acid composition results, we can conclude that the meat from pasture-fed Aberdeen Angus steers is apparently more beneficial for the consumer’s health than the meat from animals in a concentrate system.

When considering color results in the meat in the current investigation, it was observed that the beef from animals finished on pasture was darker (L* = 34.5 ± 2.2) compared with those finished on grains (L* = 36.2 ± 4.4). Similar results were found in other research studies [4,36,48,68,69,70,71]. This darker color was explained [69] by a higher myoglobin content in the pasture-fed animals. In our case, this cannot be the explanation because no differences in heme iron content were observed between the systems heme iron content were observed. Another hypothesis is that the animals produced via an extensive system perform more physical activity compared with animals in a confined system [72]. Physical exercise elevates the concentration of reactive oxygen species (ROS) in vivo, causing oxidative damage to tissues, including muscle. During the conversion of muscle into meat, biochemical events favor oxidation, and the intermediate free radicals that are generated are similar to ROS and contribute to oxymyoglobin oxidation, causing a loss of color in the meat [73]. Another hypothesis is that animals produced on pastures are older than those in the concentrate system, which is reflected in darker meat, but in the present study, the age difference between the systems was 30 days to reach similar pre-slaughter weights. 

The greater redness observed in the meat from the concentrate-fed animals was in opposition to what was found by the authors of [8] in *Longissimus thoracis* muscle from British and Zebu cross steers at days 10 and 13 of refrigerated storage, the authors of Ref. [4] in *Psoas major* steaks from crossbreed steers at 7 days of retail display, and the authors of [74] in *Longissimus thoracis et lumborum* muscle of steers over 10 days of display. However, the authors of [51,75] did not find a feeding system effect (pasture vs. grain) in the a* parameter in the *Longissimus thoracis et lumborum* muscle of steers. In general, consumers prefer a bright red color in beef because it indicates freshness. This determines purchasing decisions at retail points, so the results of the present research favor meat from concentrate-fed animals. Hence, the fact that no differences were found for heme iron content between meat from different feeding systems in the current investigation cannot explain the differences in redness. Regarding the heme iron content results found in other research studies, no differences between feeding systems were observed in *Longissimus thoracis et lumborum* and *Psoas major* muscles in the Charolais breed and crossbreed (British × Indicus), respectively [4,75].

Yellowness in meat, which is indicated by the b* parameter, presented no feeding system effect. This was in accordance with the results of [51], in which the *Longissimus thoracis et lumborum* muscle of Hereford steers was investigated, the results in Ref. [76], in which the *Longissimus thoracis et lumborum* muscle of beef was investigated, and the results in [4], in which the *Psoas major* of steers was investigated. However, in [8], steaks from pasture-fed steers were significantly yellower when compared to grain steaks.

During the time of display, some variations can be seen in the L*, a*, and b* parameters. As seen in Figure 3a, lightness is more stable in the *Semimembranosus* muscles from pasture-produced Aberdeen Angus steers than in the muscles from concentrate-fed steers during the refrigerated display because fewer variations can be seen. The decline in the a* value observed in meat from both systems indicates a loss of redness, which indicates myoglobin oxidation and was also observed in several publications [4,8,41,43,73,75,77] in beef under retail display conditions. Even though a decrease in heme iron content in the meat was expected during the nine days of display in a refrigerated showcase because meat loses fluids and consequently heme iron and myoglobin [77], the heme iron content did not show differences between the days of the display in the present research. When comparing our b* results with other works, the authors of Ref. [75] found that b* values remained stable during the first 2 days and decreased after 6 days of storage, and in [4], in animals from pasture and concentrate systems, this decrease was observed on day 6 of display but the values stayed the same until day 9. In addition, a decrease in b value was also observed in meat with increasing days of storage [8]. These variations in a* and b* values found in the current research during the time of display were reflected in variations in the tone (hue angle) and saturation (chroma) of the meat’s color. The hue angle is a good indicator of the stability of meat in retail display [37]. It seems that the loss of redness in meat during the time of display influences the tone, which changed to an orange tone, indicated by a hue value near 50, at the end of the display. Color saturation, which diminishes with the time of display, also seems to be affected by the loss of redness. Indeed, a decrease in chroma under retail display conditions is expected to occur and could be considered an indicator of metmyoglobin accumulation on the surface of the meat [37]. In [42], a decrease in chroma values during refrigerated storage was also observed in the *Longissimus lumborum* muscle of beef.

## 5. Conclusions

When the *Semimembranosus* muscle was submitted to a refrigerated retail display process which simulated commercial conditions, the meat from pasture-fed Aberdeen Angus steers was more stable from an oxidative perspective, and the oxidation processes started later than in the meat from concentrate-fed steers. This fact occurred in lipid oxidation, even though the meat from pasture-fed animals presented higher contents of two fatty acids which are very sensitive to oxidation, as α-linolenic acid and DHA. Therefore, this result could probably be explained by the higher contents of antioxidant compounds found in the meat from pasture-fed steers, such as β-carotene and α-tocopherol, both of which were determined in the present investigation. This allows the meat from animals that are pasture-fed to achieve a longer shelf life. In addition, pasture feeding may improve the lipid profile of the beef for consumers, highlighting the greater n-3 content in intramuscular fat. Concerning color stability in the *Semimembranosus* muscle during the display time, the concentrate system seems to be better because the meat had a higher redness value, which is preferred by consumers at retail points, and presented a significant increment in tone afterward than the meat from pasture-fed animals. This study generates new information about the oxidative and color stability of meat under retail display conditions in another muscle which is not studied as frequently; this muscle is frequently consumed as steak in South America and is the muscle from which carpaccio, an export product of Uruguay, is made. 

## Figures and Tables

**Figure 1 animals-13-02972-f001:**
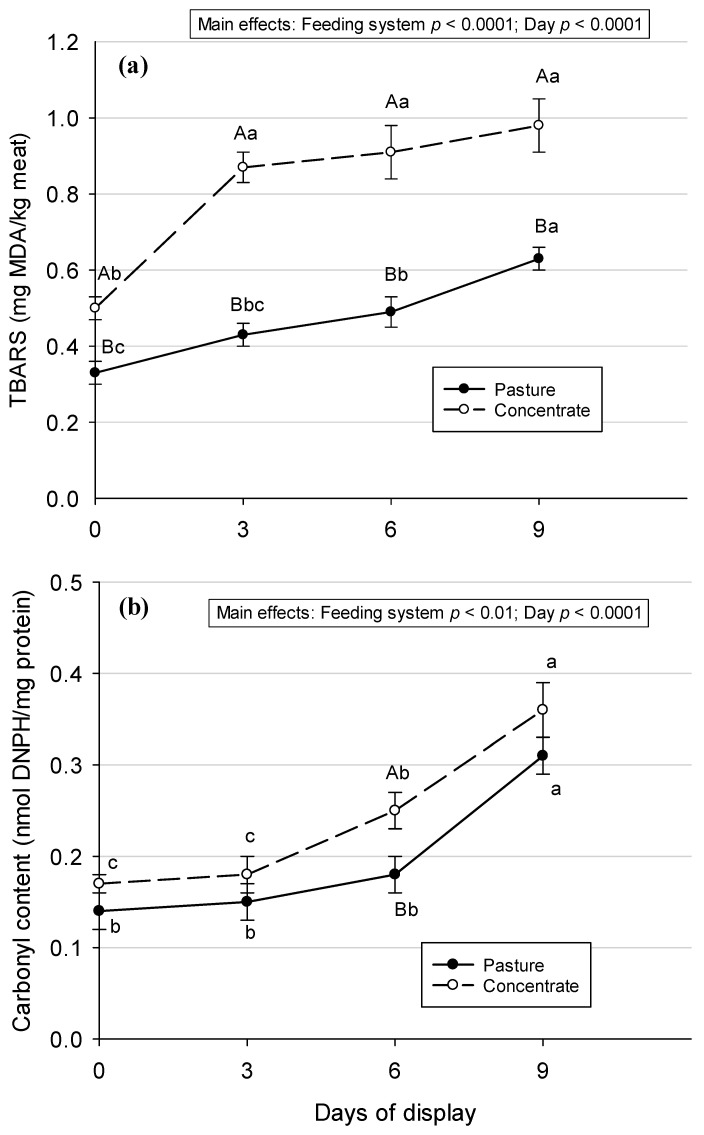
Lipid oxidation (**a**) and protein oxidation (**b**) before and during retail display in *Semimembranosus* muscles from Aberdeen Angus steers fed pasture or concentrate. Values are means ± SEM (n = 10). Different capital letters show significant differences between feeding systems on each day of the display (*p* < 0.05). Different lowercase letters show significant differences between days of display in each feeding system (*p* < 0.05).

**Figure 2 animals-13-02972-f002:**
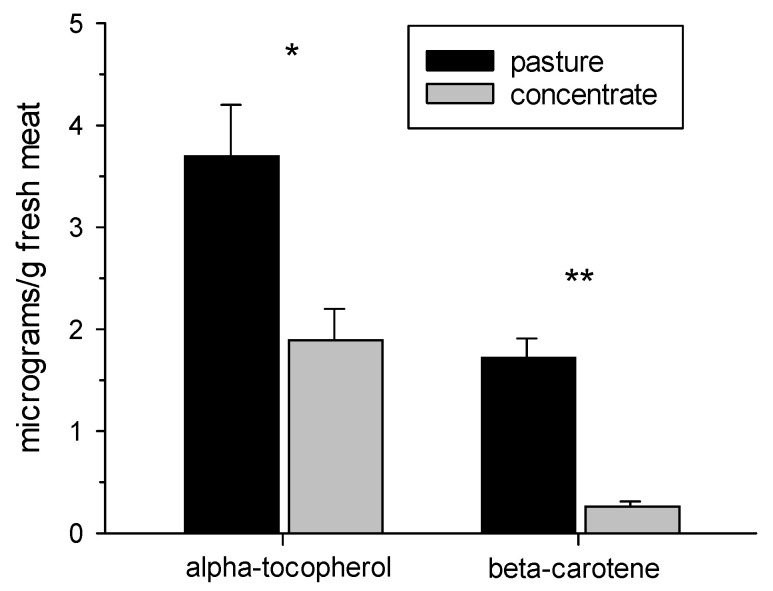
α-tocopherol and β-carotene contents (μg/g fresh meat) in *Semimembranosus* muscles from pasture- or concentrate-fed Aberdeen Angus steers prior to being placed in a retail display. Values are means ± SEM (n = 6). Significant differences between feeding systems are indicated as * (*p* < 0.05) or ** (*p* < 0.0001).

**Figure 3 animals-13-02972-f003:**
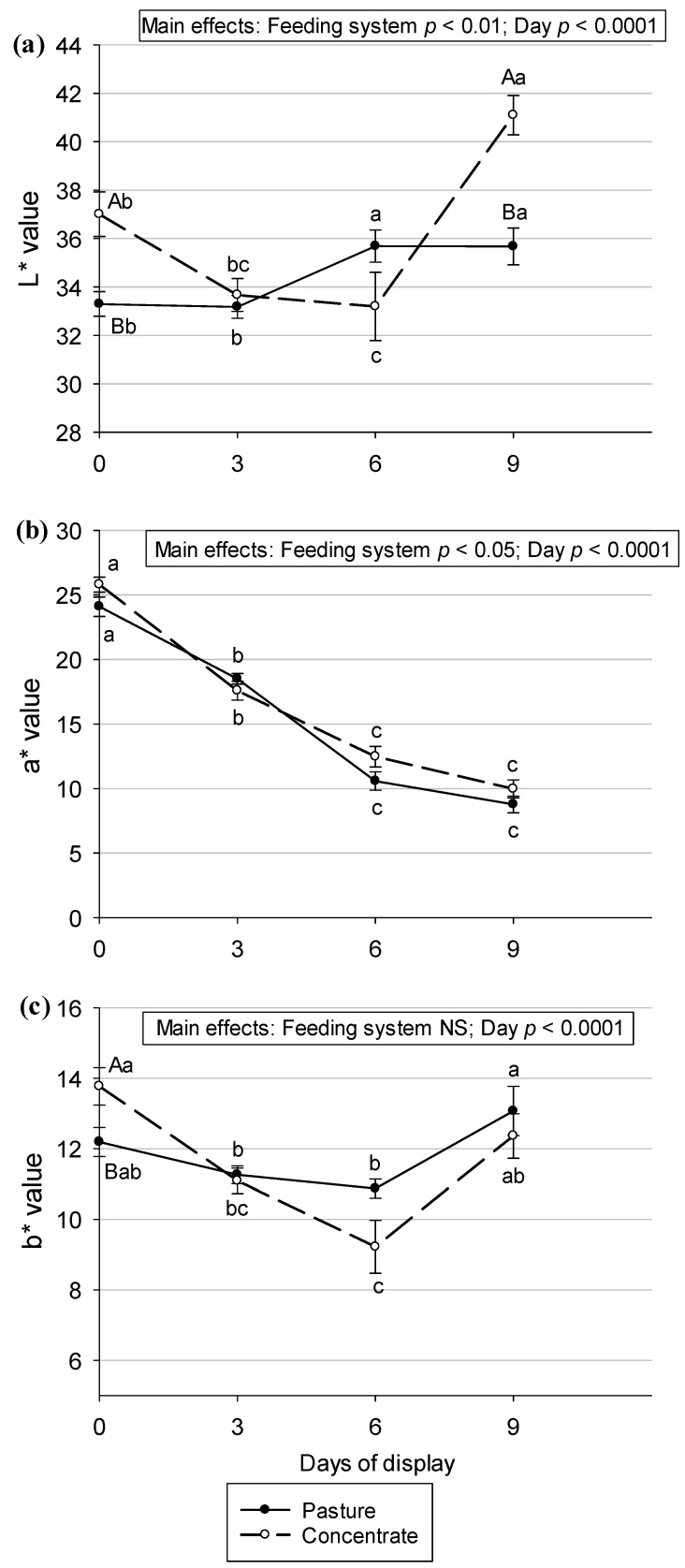
Color parameters, (**a**) L* value, (**b**) a* value, and (**c**) b* value, before and during retail display in *Semimembranosus* muscle of pasture- or concentrate-fed Aberdeen Angus steers. Values are means ± SEM (n = 10). Different capital letters show significant differences between feeding systems on each day of the display (*p* < 0.05). Different lowercase letters show significant differences between days of display in each feeding system (*p* < 0.05). NS: not significant.

**Figure 4 animals-13-02972-f004:**
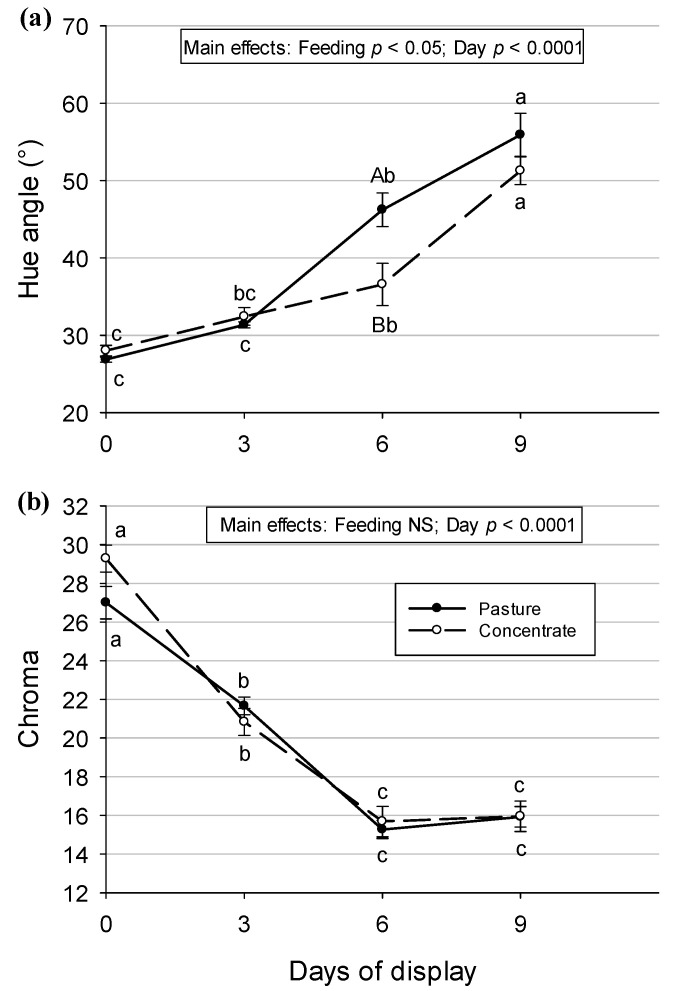
(**a**) Hue angle (°) and (**b**) chroma values before and during retail display in *Semimembranosus* muscle of pasture- or concentrate-fed Aberdeen Angus steers. Values are means ± SEM (n = 10). Different capital letters show significant differences between feeding systems on each day of the display (*p* < 0.05). Different lowercase letters show significant differences between days of display in each feeding system (*p* < 0.05). NS: not significant.

**Figure 5 animals-13-02972-f005:**
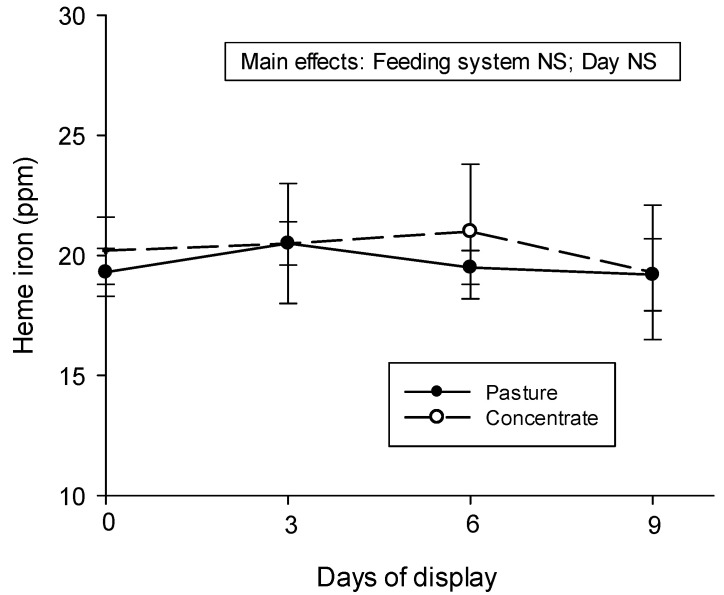
Heme iron content (ppm) in *Semimembranosus* muscle from pasture- or concentrate-fed Aberdeen Angus steers before and during the retail display. Values are means ± SEM (n = 6).

**Table 1 animals-13-02972-t001:** Total intramuscular fat content (%) and fatty acid composition (g/100 g fatty acids) in *Semimembranosus* muscles of pasture- or concentrate-fed Aberdeen Angus steers prior to retail display.

	Pasture	Concentrate	*p*-Value
% Fat	1.78 ± 0.15b	4.52 ± 0.46a	<0.0001
C12:0	0.13 ± 0.04	0.08 ± 0.03	NS
C14:0	3.46 ± 0.71	3.10 ± 0.85	NS
C15:0i	0.25 ± 0.04a	0.12 ± 0.03b	0.01
C15:0ai	0.26 ± 0.03	0.17 ± 0.04	NS
C14:1	0.60 ± 0.12	0.59 ± 0.17	NS
C15:0	0.72 ± 0.10	0.51 ± 0.11	NS
C16:0i	0.20 ± 0.03a	0.10 ± 0.01b	<0.05
C16:0	31.85 ± 2.71	28.75 ± 2.45	NS
C16:1	4.16 ± 0.38	4.22 ± 0.35	NS
C17:0	1.10 ± 0.04b	1.27 ± 0.05a	0.01
C17:1	0.74 ± 0.24	0.94 ± 0.07	NS
C18:0	12.12 ± 1.25	12.53 ± 1.08	NS
C18:1	34.46 ± 2.91	41.53 ± 2.57	NS
C18:2 n-6	2.84 ± 0.36	3.07 ± 0.22	NS
C20:1	0.13 ± 0.04	0.21 ± 0.08	NS
C18:3 n-3	0.61 ± 0.04a	0.18 ± 0.03b	0.001
CLA	0.31 ± 0.03	0.31 ± 0.09	NS
C20:3 n-3	0.14 ± 0.04a	0.03 ± 0.01b	0.01
C20:3 n-6	0.34 ± 0.10a	0.11 ± 0.02b	0.05
C20:4 n-6	0.30 ± 0.12	0.21 ± 0.06	NS
EPA n-3	0.04 ± 0.05	0.01 ± 0.01	NS
DPA n-3	0.13 ± 0.08	0.04 ± 0.02	NS
DHA n-3	0.41 ± 0.06a	0.08 ± 0.07b	<0.01
Others	4.73 ± 1.37	1.92 ± 0.41	……
SFA	50.09 ± 2.54	46.64 ± 2.39	NS
MUFA	40.08 ± 2.68b	47.49 ± 2.19a	<0.05
PUFA	5.13 ± 0.62	4.02 ± 0.49	NS
Σn-6	3.79 ± 0.41	3.69 ± 0.37	NS
Σn-3	1.33 ± 0.24a	0.33 ± 0.13b	<0.001
n-6/n-3	2.85 ± 0.35b	11.18 ± 3.96a	<0.05
PUFA/SFA	0.10 ± 0.02	0.09 ± 0.02	NS

Values are means ± SEM (n = 10). Different letters in the same row show significant differences between feeding systems (*p* < 0.05). CLA: conjugated linoleic acid; EPA: eicosapentaenoic acid; DPA: docosapentaenoic acid; DHA: docosahexaenoic acid; SFA: saturated fatty acid; MUFA: monounsaturated fatty acid; PUFA: polyunsaturated fatty acid; NS: not significant. i: iso; ai: anteiso.

## Data Availability

The data presented in this study are part of the PhD thesis of Alejandra Terevinto within the framework of a Doctorate in Agricultural Sciences (Facultad de Agronomía, Universidad de la República, Uruguay) and are available at: https://hdl.handle.net/20.500.12008/29294 (accessed on 24 July 2023).

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
