# Peer review of "The Oxidative and Color Stability of Beef from Steers Fed Pasture or Concentrate during Retail Display"

_animals, 2023, doi:10.3390/ani13182972_

Round 1

Reviewer 1 Report

Previous publications have already shown that finishing cattle on high quality pasture, supplemented or not with concentrate, provides meat with better nutritional and functional quality for humans and with longer shelf life (greater oxidative stability). Therefore, in this context, this research is incremental.

However, the results contribute to reinforce this concept and add information for two beef production systems used in Uruguay, in intercropped pasture and in confinement based on conserved forage plants (silage) as a source of roughage with the inclusion of concentrated feed.

For a better characterization of the "concentrated" system, I believe it is necessary to describe the quantitative and qualitative participation of the ingredients used, so that the reader can accurately understand the type of diet used. The diet of the intensive system (confinement) was labeled as concentrate, but it includes the participation of preserved forages.

Therefore, I recommend that the quantitative and qualitative characteristics of the experimental diets are presented in material and methods and that these are also explored in the results and discussion chapters and in the conclusions.

Author Response

The information requested can´t be included in the manuscript because we did not formulate the rations. We did not have access to the percentage of ingredients used in the concentrate-based diet because farm producers were in charge of formulating the rations, and we dealt directly with the meat processing plant to select the meat cuts. All the information we have regarding the experimental diets is mentioned in the manuscript and was taken from traceability and information given by the meat processing plant. New information was included in the materials and methods section to complement it. The Materials & methods section was improved for a better understanding.

Reviewer 2 Report

The authors used animals prevenient from intensive and extensive finishing systems and studied the oxidative and color stability of the Semimembranosus, also fatty acid composition.

“Beef meat” – title: beef is the meat of bovine

What is the question the authors want to answer? The results are a little bit blocked and maybe once it is clear it could read mor smooth.

The statistical analysis topic needs more information on how the analysis was performed. What is the experimental design? What is the experimental unity? If animal or muscle is the experimental unity, maybe some data should be analyzed as repeated measurements.

Does beef from grass-fed cattle have more oxidative stability or does it have less fat, hence less lipid oxidation, which affects protein oxidation and color stability? Differences in the finishing system affect the fat content and if one treatment has more fat, it will have more lipid oxidation. Does it make the grass-fed more stable? If you adjust your statistical model for fat content, you can see how they differ.

In fact, muscles from grass-fed animals have more b-carotene, but there are many other antioxidant components in the muscle cell that need to be considered when talking about oxidative capacity.

L* and a* values showed a higher discoloration in grass-fed beef, which doesn’t match the higher antioxidant stability of the grass-fed as the authors suggested. 

Author Response

The words “Beef meat” were changed for “beef” in the title and in the rest of the manuscript. It is highlighted in the revised manuscript.

The question authors want to answer is whether a muscle not so much studied and usually seen at retail points in the form of steaks in Uruguay, coming from a pasture system is more stable than the one coming from a concentrate system, when put in retail display conditions. Also, if antioxidants present in meat or fatty acid composition may explain these differences.  Results were separated from the Discussion in the revised manuscript and hope that they are better understood now.

The Statistical analysis section was improved. More details were included. The experimental unit was muscle. This was added in the manuscript. Some data was analyzed as repeated measures.

The fact that fat content can affect the oxidative stability in meat was considered, and it was mentioned in the Discussion section. In general, the concentrate feeding system produces meat with a higher intramuscular fat content compared to a pasture feeding system, and having a higher fat content makes meat more likely to have a higher lipid oxidation rate.

Beta-carotene and alfa-tocopherol were the only antioxidants measured in meat in this research, but based on the bibliography consulted alfa-tocopherol has the greatest antioxidant power in meat retarding the oxidative processes, and its content is really greater in meat from pasture-fed animals. Of course, there are other antioxidant compounds present in muscle cells like antioxidant enzymes (SOD, catalase, and GPx) but these were not part of the present study.

Lightness (L* value) was lower in meat from pasture-fed animals indicating that is darker than meat coming from concentrate-fed animals. Possible explanations were mentioned in the manuscript. When observing the L* value during the retail display, more variations can be seen in meat from concentrate-fed steers, hence we can say that meat from pasture-fed steers is more stable regarding luminosity.

Redness (a* value) was lower in meat from pasture-fed animals compared to meat from concentrate-fed animals during the retail display. In this sense, the discussion of color stability was changed in favor of meat from concentrate-fed steers in the revised manuscript. This was highlighted.

Reviewer 3 Report

Oxidative and color stability of beef meat from steers finished on pasture or concentrate during retail display.

I have read the paper, and found some also need improve.

In this paper, Similar results were found in other research........, Please explain the innovation of this article.

The color is closely related to the state of myoglobin. Why was the content of myoglobin, oxygenated myoglobin, and ferrimyoglobin not measured in the article?

The title need improve.

Abstract need improve, because it is a description of the results, not a summary.

L60-66 The sentences need improve.

Statistical analysis need improve.

L183-191 The sentences need improve.

L209 In the present investigation, these results could be due to a higher........Why? Please explain!

L226-231 The sentences need improve.

L239-243 The sentences need improve.

Conclusion need improve.

Oxidative and color stability of beef meat from steers finished on pasture or concentrate during retail display.

I have read the paper, and found some also need improve.

In this paper, Similar results were found in other research........, Please explain the innovation of this article.

The color is closely related to the state of myoglobin. Why was the content of myoglobin, oxygenated myoglobin, and ferrimyoglobin not measured in the article?

The title need improve.

Abstract need improve, because it is a description of the results, not a summary.

L60-66 The sentences need improve.

Statistical analysis need improve.

L183-191 The sentences need improve.

L209 In the present investigation, these results could be due to a higher........Why? Please explain!

L226-231 The sentences need improve.

L239-243 The sentences need improve.

Conclusion need improve.

Author Response

With this investigation we tried to generate new information regarding a muscle not so much studied. This muscle is highly consumed in Uruguay because it´s lean and cheaper than other meat cuts and is commonly seen in the form of steaks in refrigerated showcases at retail points. This was included in the revised manuscript in the Materials & methods section.

Besides, it´s common to see meat exposed at retail points for consumers´ purchase, and it´s important for them and the meat industry to know what meat, from pasture or from concentrate, has a longer shelf life, because it can cause economic losses.

Also, to our knowledge, it´s the first time that β-carotene is measured in meat produced in Uruguay, and the second time for α-tocopherol. This was added in the Discussion section.

Furthermore. results of this research reaffirm the fact of a greater content of β-carotene and α-tocopherol, and a greater oxidative stability in meat from animals produced in pasture found in other international research, as well as a healthier fatty acid profile of intramuscular fat.

Regarding the question of why the content of myoglobin, oxygenated myoglobin, and ferrimyoglobin were not measured in the article, the authors answer that in this sense, heme iron content was measured because it´s associated with changes in meat color.

Title was improved.

The authors do not agree with the comment about the abstract. In addition to the results, a short introduction, the objective of the work, the materials and methods used, and the conclusions are also mentioned in the abstract.

L60-66 The sentences need improve. 

Answer: Connectors were added to enlace the sentences, and a part of the sentence that repeated the same fact “β-carotene is a fat-soluble antioxidant” was removed, so as to improve the manuscript.

The experimental unit (muscle) was added in the Statistical analysis section. More details were included.

R: L183-191 The sentences need improve. 

Answer: The sentences were improved adding connectors. Please let us know if this is enough.

L209 In the present investigation, these results could be due to a higher........Why? Please explain! 

Answer: The order of the sentences was changed for a better understanding of the explanation of the results, but the explanation was described in the manuscript.

R: L226-231 The sentences need improve.

Answer: Please make a suggestion on how it can be improved.

R: L239-243 The sentences need improve.

Answer: This sentence was changed for a better understanding.

R: Conclusion need improve.

Answer: Some changes and more explanations were added in the Conclusion section as suggested.

Reviewer 4 Report

The manuscript is within the scope of the Animals journal and should be of interest to its readership. Authors need to highlight what the paper brings as a novelty in food science. Moreover, authors must to state the limitation of this study at the end of Discussion section. Please, see my comments and suggestions attached.

I suggest that the manuscript could benefit from a more comprehensive grammar revision by a professional English language expert. Although the grammatical quality of the English was sufficient to allow for a full interpretation of the text, it needs some improvement to be published. 

Author Response

With this investigation we tried to generate new information regarding a muscle not so much studied. This muscle is highly consumed in Uruguay because it´s lean and cheaper than other meat cuts and is commonly seen in the form of steaks in refrigerated showcases at retail points. This was included in the revised manuscript in the Materials & methods section.

Besides, it´s common to see meat exposed at retail points for consumers´ purchase, and it´s important for them and the meat industry to know what meat, from pasture or from concentrate, has a longer shelf life, because it can cause economic losses.

Also, to our knowledge, it´s the first time that β-carotene is measured in meat produced in Uruguay, and the second time for α-tocopherol. This was added in the Discussion section.

Furthermore. results of this research reaffirm the fact of a greater content of β-carotene and α-tocopherol, and a greater oxidative stability in meat from animals produced in pasture found in other international research, as well as a healthier fatty acid profile of intramuscular fat.

The limitation raised by the reviewer regarding the number of animals used in each diet treatment in this study was not a concern for the authors. Two very similar works published in Meat Science considered n= 5 (Insani et al., 2008) and n=8 (Humada et al., 2014). In other research, also published in Meat Science, on meat quality, an n=8 (Gatellier et al., 2001) and n=14 and n=18 (Realini et al., 2004) were considered.

In response to the comments and suggestions made in the manuscript, the authors want to clarify some points.

Line 24 – longissimus muscle was suggested

We know that the Longissimus dorsi is a reference muscle in meat quality research, but we wanted to generate new information regarding another not so much studied muscle, as it is the semimembranosus. This muscle is part of a cut (inisde round) that is commonly seen in refrigerated showcases at retail points in Uruguay in the form of steaks, and is more affordable for consumers compared to other more expensive cuts. 

Line 38 – ageing is not the same as refrigerated storage. Ageing is a process where meat is vacuum packaged and stored in the dark at 1-2°C for a set time to improve tenderness, and refrigerated storage can be with a vacuum or MAP package, with a PVC film, exposed to light, or not.

Results and discussion were separated.

Results of Hue and chroma were included in the manuscript.

The results of beta-carotene and alpha-tocophol reported in tables were changed to figures.

Conclusions were improved as suggested.

The manuscript was subjected to a grammar checker as suggested.

Round 2

Reviewer 2 Report

The authors addressed my comments. 

Author Response

Thank you very much for your comments. Minimal corrections were made in this second revision as suggested by other reviewer. Corrections are highlighted in the manuscript.

Reviewer 3 Report

Oxidative and color stability of beef from steers fed pasture or concentrate during retail display

I have read the paper, the authors have good reviewed, and found some need improve.

The language also need improve. Such as P<0.05 or P < 0.0001?

Figure or Fig.? .............................

Oxidative and color stability of beef from steers fed pasture or concentrate during retail display

I have read the paper, the authors have good reviewed, and found some need improve.

The language also need improve. Such as P<0.05 or P < 0.0001?

Figure or Fig.? .............................

Author Response

Thank you very much for your comments. The corrections suggested were highlighted in the manuscript.

Reviewer 4 Report

Authors have improved the manuscript following my suggestions. The manuscript can be now accept for publication.   

Author Response

Thank you very much for your comments. Minimal corrections were made in this second revision as suggested by other reviewer. Corrections were highlighted in the manuscript.